# Simultaneous sensing and imaging of individual biomolecular complexes enabled by modular DNA–protein coupling

Mario J. Avellaneda [1], Eline J. Koers[1], David P. Minde [1,3], Vanda Sunderlikova[1] & Sander J. Tans[1,2]✉

Many proteins form dynamic complexes with DNA, RNA, and other proteins, which often involves protein conformational changes that are key to function. Yet, methods to probe these critical dynamics are scarce. Here we combine optical tweezers with fluorescence imaging to simultaneously monitor the conformation of individual proteins and their binding to partner proteins. Central is a protein–DNA coupling strategy, which uses exonuclease digestion and partial re-synthesis to generate DNA overhangs of different lengths, and ligation to oligo-labeled proteins. It provides up to 40 times higher coupling yields than existing protocols and enables new fluorescence-tweezers assays, which require particularly long and strong DNA handles. We demonstrate the approach by detecting the emission of a tethered fluorescent protein and of a molecular chaperone (trigger factor) complexed with its client. We conjecture that our strategy will be an important tool to study conformational dynamics within larger biomolecular complexes.

[1] AMOLF, Amsterdam 1098XG, The Netherlands. [2] Department of Bionanoscience, Kavli Institute of Nanoscience Delft, Delft University of Technology, Van der Maasweg 9, 2629 HZ Delft, The Netherlands. [3]Present address: Cambridge Centre for Proteomics, University of Cambridge, Cambridge CB2 1QR, UK ✉email: s.tans@amolf.nl

It is well known that conformational change is central to protein function and folding. At the same time, binding partners that both depend on, and affect these conformational changes, are crucial within the cellular context[1]. Indeed, in cells, proteins typically function transiently within functional complexes[2], respond to ligand binding in signaling pathways[3], regulate gene activity[4], and interact with the protein homeostasis machinery from synthesis to degradation[5,6]. Yet, studying this interplay between protein interactions and conformational change is challenging. Advances in cryogenic electron microscopy, nuclear magnetic resonance, and X-ray crystallography are revealing protein complexes in increasing structural detail but do not address the conformational and binding dynamics that play a central role in their function[7–9].

In the last decades, single-molecule force spectroscopy has provided key insights into diverse molecular systems and mechanisms[10]. In this approach, forces and displacements are measured on molecules tethered between trapped beads, atomic force microscopy cantilevers, and surfaces. Recently, force spectroscopy has been combined with imaging techniques such as wide-field and confocal fluorescence microscopy, Förster resonance energy transfer (FRET), or stimulated emission depletion[11–15]. These approaches have so far mainly been applied to study the binding of partners and other ligands to DNA, with DNA strands being tethered to allow mechanical manipulation, while DNA-binding partners are detected using fluorescence imaging[16–20].

Protein–protein interactions have been extensively studied using force spectroscopy alone, including peptide translocases[21–23], molecular chaperones[24–29], crosslinking proteins such as catch bonds[30,31], molecular motors[32,33], or protein assembly[34]. Simultaneous fluorescence imaging provides a powerful tool to better understand protein–protein complexes. Direct visualization of protein binding relaxes the stringent requirements for large statistical samples[26,35], because force events can then be correlated directly to the presence of bound partners. Proteins in complexes also often act synergistically and at different moments in time. Their direct imaging allows the study of causal binding and conformational events in time, whereas FRET can reveal conformational information that remains hidden with force sensing alone, e.g., within untethered proteins that are part of the complex. Finally, fluorescence imaging allows monitoring of the number of bound proteins in time, which is of direct relevance in protein assembly or oligomeric complexes.

However, dual sensing–imaging experiments on protein complexes have remained inaccessible thus far. A key challenge is to achieve efficient and strong coupling of the proteins to long DNA handles. DNA handles permit bead attachment while limiting bead–surface interactions and laser damage[36,37]. As we also show here, the combination of fluorescence imaging requires far longer DNA tethers (over 4 kbp), to limit the parasitic fluorescence from trapped beads and photobleaching caused by the trapping lasers. Additional strategies such as interlaced trapping–imaging can help mitigate some of these issues[38]. Coupling efficiency, strength, and durability of DNA handles constitute a general obstacle and often determine whether single-molecule force spectroscopy is feasible or not, even without fluorescent detection. Fluorescence and protein–protein interactions further exacerbate these issues. High forces are required to unfold proteins stabilized by bound proteins or ligands[26,39], or to quantify forces exerted by molecular motors or peptide translocases[22,23]. Moreover, complex formation can take up to hundreds of seconds, because background fluorescence limits achievable concentrations, which becomes impractical when tether durability is limiting[40].

Current approaches typically use thiol chemistry to directly attach DNA tethers to cysteine residues[41], or to first couple short DNA oligos and then hybridize longer DNA handles[42]. The former yields strong coupling but is practically limited to short tethers below 500 bp, in part due to the electrostatic repulsion of large DNA molecules[41]. The two-step method has been used for longer handles up to 3 kbp[43]. However, the involved hybridization interactions provide lower mechanical stability than the former direct coupling approach and cannot resist high forces for extended periods of time[43,44].

Here we present a new general DNA–protein coupling method for combined protein sensing and imaging. Twenty nucleotide-long oligos (anchors) are first coupled to proteins via cysteine chemistry or enzymatic reaction and then covalently ligated to DNA tethers of over 5000 bp. To generate DNA handles with ligation-compatible overhangs of any size, we use complete digestion of one of the DNA strands, followed by partial re-synthesis. This strategy provides an advance for protein force spectroscopy applications that do not use fluorescence or high forces, through increased coupling efficiency and tether durability. Moreover, it enables combined sensing–imaging and high-force applications, by efficiently generating long and stable constructs, which limit parasitic fluorescence from trapping beads and sustain DNA overstretching forces (>60 pN) during long periods of time (>10 min). To demonstrate this approach, we tether proteins between beads trapped by optical tweezers, while scanning a confocal excitation beam and detecting the fluorescence emission, which allows visualization of a single fluorescent protein and the binding of an individual chaperone to a tethered client.

## Results and discussion

**Coupling of short DNA oligos to proteins.** First, we addressed the protein–anchor coupling, which is key to overall efficiency in existing hybridization approaches[45]. Specifically, we interrogated the effect of the anchor length. Maltose-binding protein (MBP) with cysteines at both termini was incubated with a fourfold excess of maleimide-modified anchors of 20, 34, and 40 nucleotides (nt) in length (Fig. 1a), and coupling results were analyzed by SDS-polyacrylamide gel electrophoresis (Fig. 1b and Supplementary Fig. 1). For the longer 40 nt anchor, about 19% of the product corresponded to coupling of two oligos to the protein and the rest either did not couple or to one terminus only (Fig. 1d and Supplementary Fig. 1a). Decreasing the length of the anchor

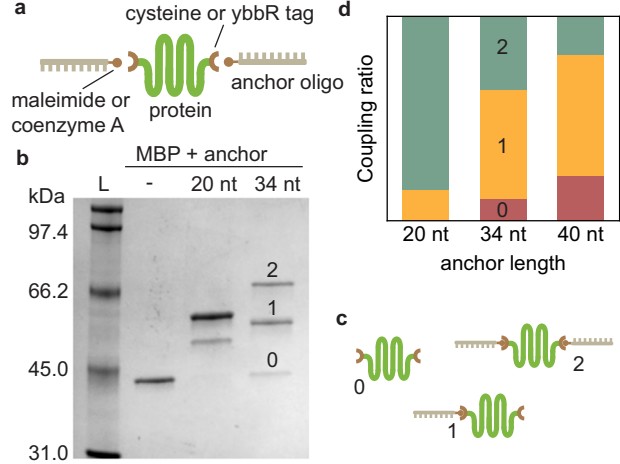

**Fig. 1 Shorter anchors provide higher protein–anchor coupling yields. a** Scheme of the anchor oligo coupling to a modified protein. **b** SDS-PAGE analysis of the coupling products. Lane L: protein ladder, next: MBP; MBP reacted with 20 nt anchors; and 34 nt anchors. **c** Possible products of the coupling reaction. **d** Coupling ratios for different anchor lengths.

resulted in a notable increase in coupling yield, with 36% anchor–protein–anchor for the 34 nt anchor and 85% for the 20 nt anchor (Fig. 1a–d and Supplementary Fig. 1a), in line with previously reported efficiencies (just below 20% for a 34 nt anchor)[45].

To study compatibility of our ligation method with other anchor-coupling chemistries, which can access a wider range of proteins that contain essential cysteines, we also tested an enzymatic reaction. We genetically introduced a ybbR tag (DSLEFIASKLA) at each terminus of YPet (a yellow fluorescent protein variant), which were then coupled to anchors modified with coenzyme A (CoA) using Sfp synthase (Sfp 4'-phospho-pantetheinyl transferase; see Methods and Supplementary Fig. 1b, c)[46]. Here we found that 27% of proteins coupled to two 20 nt anchors (Supplementary Fig. 1d). Other coupling chemistries can be used to attach the anchors to the protein of interest, including sortase-mediated reactions[47], click chemistry[48], and a range of peptide tags[49,50]. Many of these reactions are typically less efficient than cysteine chemistry and we surmise that the reduced length of the anchors used here provides higher coupling yields than previous protocols for any modification chemistry.

**Generation and coupling of ligation-compatible DNA handles.** Next, we considered the anchor-handle linkage, which is central to the mechanical stability against applied forces. Previously, oligo anchors have been hybridized to a complementary overhang of the DNA handles, generated using abasic primers[42]. Hybridization yields non-covalent linkages that can limit mechanical stability against applied force, especially for shorter anchors[44]. Therefore, existing protocols typically employ anchors of at least 34 nt to increase mechanical stability. However, as shown above (Fig. 1), such longer anchors come at the cost of lower anchor–protein–anchor coupling efficiencies. This tradeoff may, in principle, be overcome by DNA ligation, if the latter proves to be efficient, as one can then use shorter efficiently coupling anchors while also achieving high strength. However, the abasic site used in current methods hinders efficient ligation[51]. Restriction enzymes can generate ligation-compatible overhangs, but they are limited to 4–6 nt and yield dual-handle coupling efficiencies lower than 5% even for DNA molecules that are too short for the present purpose (<400 bp)[52]. We developed a strategy consisting of three consecutive rapid enzymatic treatments to generate DNA overhangs unrestricted in length that can be covalently ligated to the coupled anchors (Fig. 2).

First, a 1333 bp-long DNA template was generated using a phosphorylated forward primer and a functionalized reverse primer for attachment to the bead or surface (Figs. 2a1 and Fig. 2c, lane 1). Here we chose biotin and digoxigenin. The product was digested with λ exonuclease (Fig. 2a2 and Fig. 2c, lane 2) and the remaining functionalized single-stranded DNA (ssDNA) strand was then used for a partial re-synthesis, where the primer sequence is complementary to an inner segment of the strand, starting where the anchor-complementary sequence finishes (Fig. 2a3). To preserve the overhang, we used Deep Vent (exo-) polymerase, which lacks 3' → 5' proofreading exonuclease activity[53] (Figs. 2a4 and Fig. 2c, lane 4). The overhang length can be varied with this approach by the appropriate primer choice. More importantly, the generated overhang allows covalent DNA ligation and permits to use shorter, more coupling-efficient anchors without limiting the resistance of the tethers.

The anchor–protein–anchor construct was ligated to the 1333 bp-long tethers (ratio 1:1:1) with T4 ligase (Fig. 2b), and an agarose gel electrophoresis analysis showed that 45% of the handles were ligated together into a complex twice the size (Fig. 2c, lane 6). Consistently, in the the absence of

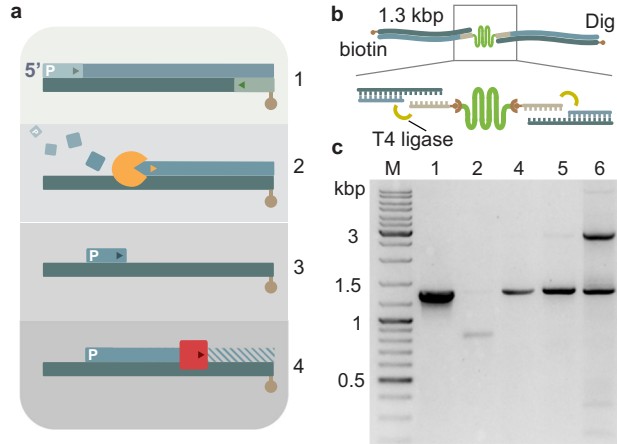

**Fig. 2 DNA handle generation and attachment. a** Strategy for the overhang generation. 1 Initial PCR amplification of template DNA using phosphorylated and functionalized primers. 2 The λ exonuclease (orange) digestion of the phosphorylated strand. 3 Tuning of the overhang length by selection of the appropriate primer. 4 Partial strand re-synthesis using Deep Vent (exo-) (magenta) that leaves the overhang intact (not abasic) for ligation. **b** Handle attachment scheme. The yellow arcs represent T4 ligation. **c** Agarose gel electrophoresis analysis of the tethering. Lane M: DNA ladder. Lane 1: initial 1333 bp template. Lane 2: λ exonuclease digestion, with a lower band at around 700 bp, indicating successful digestion (dim signal because of ssDNA). Lane 4: partial re-synthesis showing that strand extension is complete (band is back at 1300 bp). Lane 5: ligation of overhang DNA only (no anchor–MBP–anchor), indicating that unspecific ligation between handles is negligible. Lane 6: ligation of overhang DNA with anchor–MBP–anchor, showing an upper band at 2600 bp. The numbering 1–4 corresponds to **a**.

anchor–protein–anchor, almost none of the handles were ligated (1%; Fig. 2c, lane 5). A high-temperature treatment in the presence of free anchor confirmed the handles were indeed ligated, as most remained linked, in contrast to their detachment when ligation was not performed (Supplementary Fig. 2).

**Mechanical stability characterization with optical tweezers.** To show the improved mechanical stability provided by ligation with respect to existing hybridization protocols, we linked the construct between functionalized polystyrene beads with optical tweezers (Fig. 3a). Resulting force-extension curves for MBP showed the characteristic unfolding pattern in two steps (Fig. 3b)[25]. We quantified tether strength by recording the maximum tensions they reached without breaking when slowly ramping up the applied force (Supplementary Fig. 3a). All tethers that were generated by hybridization only, without ligation, were found to break below 47 pN (Fig. 3c; $N = 33$), close to the predicted shearing force for our anchors (45 pN)[44]. In contrast, the majority of ligated tethers (71%, $N = 28$) could be pulled up to the DNA over-stretching regime—above 60 pN[54,55]—without rupturing for multiple cycles ($N_{cycles} = 106$), thus demonstrating the improved mechanical stability provided by ligation (Fig. 3c). We also measured tether lifetimes at 30 pN, well below the expected shearing force (Supplementary Fig. 3b). Ligation yielded a remarkable lifetime improvement of two orders of magnitude, to over 100 s ($N_+ = 21$, $N_- = 15$; Fig. 3d). These data underscored the poor mechanical stability provided by short hybridized anchors even at low forces and the utility of the exonuclease approach to overcome these limitations and enable strong and efficient ligation. For experiments where longer lifetimes are required, one may replace the digoxigenin connection by another link[49,56,57]. One may also

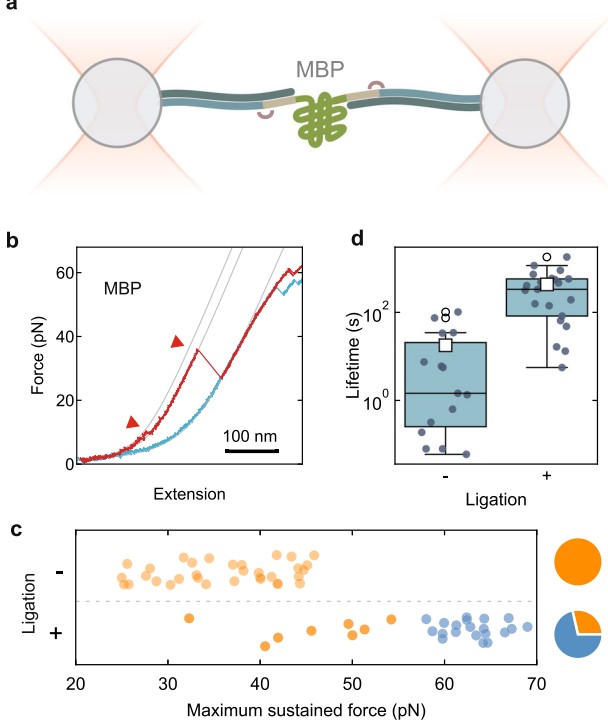

**Fig. 3 Ligation provides higher mechanical stability. a** MBP tethered with DNA between two beads trapped with optical tweezers. The arcs indicate ligation. **b** Force-extension curve of MBP with 1300 kb handles showing the characteristic two-step unfolding pattern (red triangles) and the DNA overstretching regime above 60 pN (red: pulling, blue: relaxing, gray: worm-like chain fitting curves). **c** Distribution of maximum reached force for non-ligated and ligated tethers (red indicates broken, green unbroken tethers). Pie charts show the distributions of broken and unbroken molecules. **d** Tether lifetime at 30 pN (well below the predicted rupture force of the anchors) without and with ligation (scale is logarithmic).

use biotin connections on both ends, and then tether the construct between the beads by means of high-speed laminar flow to avoid connecting both ends to one bead[58]. We also note that the DNA handles indicate a limit of about 65 pN. Although higher forces can be used to obtain unfolded proteins, it is then difficult to discriminate folding transitions from DNA unwinding events[58].

**Imaging of single fluorescent proteins using long DNA handles**. Next, we tested whether our tethers allowed simultaneous fluorescence detection and mechanical sensing. We tethered the fluorescent protein YPet to trapped beads, while scanning a confocal excitation beam along the DNA–protein–DNA construct and beads (Fig. 4a). Resulting kymographs showed significant parasitic autofluorescence signals emanating from the beads, several hundred nanometers beyond their surfaces, thus obscuring the relevant signal from the tethered YPet (Fig. 4b). To overcome this issue, we generated even longer handles of 5 kbp each using our protocol and found that they could also be ligated efficiently to anchor–protein–anchor constructs despite their increased length (35%; Supplementary Fig. 4). The key region between the beads now showed a minimal background photon count, indicating a lack of bead parasitic signals (Fig. 4c). After establishing a single tether, a fluorescent spot could now be detected between the beads, indicating the presence of active YPet, as was previously reported for green fluorescent protein[43] (Fig. 4c). The low emission (here about 2–3 photons per scanline; Supplementary Fig. 5c) highlights the importance of the spacing

provided by the longer handles. In addition, YPet remains folded even at high forces (>45 pN), which our tethers resisted for tens of seconds (Supplementary Fig. 5a, b). In contrast, previous studies on stable proteins using the hybridization approach were limited to applying high forces (~45 pN) only briefly (~50 ms), owing to the risk of tether rupture[43]. Ligated tethers thus are useful to explore a wide range of conformational states and timescales[25,26].

**Monitoring of trigger factor binding to MBP**. Finally, we aimed to detect the binding dynamics of a chaperone–substrate complex. We added fluorescently labeled trigger factor, a key and abundant *Escherichia coli* chaperone, in solution (Fig. 4d). The parasitic signal from the beads, attached to a handle–MBP–handle construct, was now even stronger due to trigger factor binding to the bead surfaces. Here, the 5 kbp-long handles provided sufficient distance to overcome this issue (Supplementary Fig. 6). We could visualize the binding of single trigger factor chaperones to MBP in real time, whereas the latter was cyclically stretched and relaxed (Fig. 4e). Binding occurred after MBP unfolding. The tether durability was important to observe the infrequent binding of trigger factor, which has both low affinity and is low in concentration to limit background fluorescence. Trigger factor remained bound for periods of time ranging from brief (<1 s; Supplementary Fig. 7) to over 10 s, timescales which are well below the fluorescent dye lifetime (see Methods and Supplementary Fig. 8). The nature of the trigger factor interaction with unfolded substrates remains incompletely understood[6] and has been suggested to involve multiple low-affinity contacts[59]. The present approach reveals that trigger factor can remain bound to both relaxed and stretched substrate chains, where the number of contacts is reduced. This direct visualization of long-term binding also explains previously reported suppression of substrate refolding by trigger factor[25].

## Discussion
In summary, here we have presented a DNA–protein tethering strategy that efficiently generates long and mechanically stable constructs, for proteins that either contain essential cysteines or not. It uses shorter ssDNA anchors compared with hybridization-only approaches, which yields higher anchor–protein–anchor coupling efficiencies, while achieving high handle–anchor coupling efficiencies and without the cost of lowered mechanical stability. These features are beneficial to experiments that use force spectroscopy at moderate forces only and enables the combination with fluorescence imaging and the application of high forces, as demonstrated by two proof-of-principle examples.

We anticipate that this combined sensing–imaging approach to study protein complexes will be applied more broadly. Our approach may also be used with other strategies to generate DNA overhangs, such as nicking enzymes[60], which have been employed for DNA–DNA coupling. Single-molecule protein sensing–imaging studies have the potential to provide new insights into functional interplay between multi-protein complex formation and protein conformation, as is for instance evidenced in the functioning of molecular chaperones[6], intrinsically disordered protein networks[61], and DNA- and RNA-binding proteins including novel homologs of the CRISPR-Cas9 complexes[62], tumor repressors[63], and steroid receptors[64], among many other systems.

## Methods
**Protein expression and purification**. MBP was modified with cysteine residues using the pET28 vector. YPet (a more stable and brighter variant of yellow fluorescent protein) was fused to MBP, to improve solubility and to enable affinity chromatography, and two ybbR tags (DSLEFIASKLA) were included at each

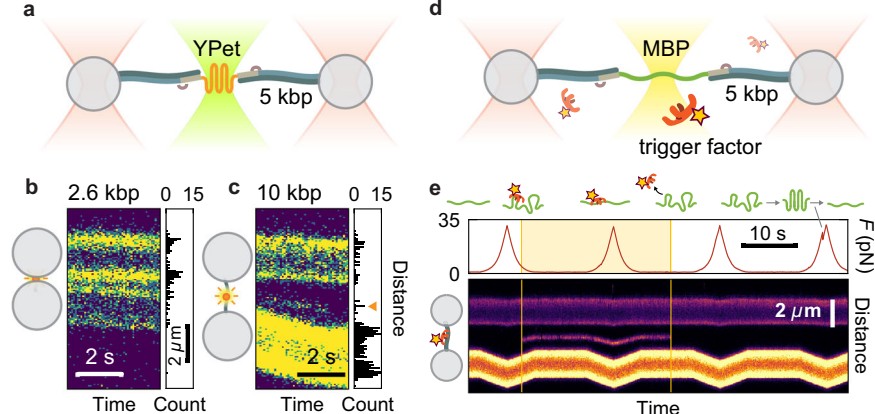

**Fig. 4 Dual monitoring of single-protein conformation and binding. a** Scheme of tethered YPet with an additional 532 nm excitation laser. **b, c** Confocal fluorescence kymographs of YPet using 1.3 kbp and 5 kbp handles, respectively, with a typical scanning line profile on the right. Parasitic fluorescence of the beads prohibits detection of protein emission when using 1.3 kb handles, whereas 5 kb tethers overcome this limitation. **d** Scheme of unfolded MBP with an additional 638 nm excitation laser and Atto647N-labeled trigger factor. **e** Force monitoring and complex formation imaging for MBP-trigger factor. Trigger factor binds to MBP after unfolding and remains bound during stretching to 35 pN.

terminus. Proteins were purified from *E. coli* BL21(DE3) cells. For overexpression, overnight cultures were diluted 1:100 in fresh lysogeny broth (LB) medium supplemented with 50 mg/l kanamycin, 0.2% glucose, and incubated under vigorous shaking at 30 °C. Expression was induced at OD600 = 0.6 by addition of 1 mM isopropyl β-D-1-thiogalactopyranoside and incubation overnight at room temperature (RT). Cells were cooled, collected by centrifugation at 5000 × *g* during 20 min, flash-frozen, and stored at −80 °C. Cell pellets were resuspended in ice-cold buffer A (50 mM potassium phosphate pH 7.5, 0.15 M NaCl, 3 mM chloramphenicol, 50 mM Glu-Arg, 10 mM Complete Protease Inhibitor Ultra from Roche, 10 mM EDTA) and lysed using a pressure homogenizer. The lysate was cleared from cell debris by centrifugation at 50,000 × *g* for 60 min and incubated with Amylose resin (New England Biolabs) previously equilibrated in buffer A for 20 min at 4 °C. The resin was washed with buffer A three times by centrifugation and bound proteins were eluted in buffer A supplemented with 20 mM maltose. Purified proteins were aliquoted, flash-frozen in liquid nitrogen, and stored at −80 °C.

**Detailed protocol for protein–anchor coupling.** Anchor oligos 5′-modified with maleimide or CoA were purchased from biomers.net and diluted in coupling buffer (Sodium Phosphate 100 mM pH 7.2, NaCl 150 mM, EDTA 10 mM) to a concentration of 300 μM or 500 μM, respectively. Purified proteins were thawed to RT and passed through a desalting column (PD-10, GE Healthcare) to get rid of reducing agents and elutants. If concentrations were below the 100 μM range, they were concentrated using an appropriate size Amicon centrifugal filter. Immediately after, they were set to the coupling reaction. For the cysteine chemistry coupling, the protein was mixed with the anchor oligos in a 1:4 ratio and incubated for 1 h at RT or overnight at 4 °C. Addition of tris(2-carboxyethyl)phosphine (TCEP) in the middle of the incubation increased the coupling yield. For the Sfp-mediated reaction, around 6 μM YbbR-modified YPet was incubated with 8 μM Sfp synthase (New England Biolabs) and 25 μM CoA-modified oligos, 50 mM Hepes pH 7.5, and 10 mM MgCl₂, in a total volume of 20 μL at RT for 1 h. Sfp synthase transfers the 4′-phosphopantetheinyl moiety of CoA to a serine residue of the ybbR tag (DSLEFIASKLA; see Supplementary Fig. 1). Excess anchor oligos were removed by affinity chromatography using amylose resin.

**Detailed protocol for overhang generation.** Initial DNA templates were generated by PCR from ~3 ng commercial pUC19 plasmid (ThermoFisher) or from pOSIP-TT (for 5 kb tethers) using Phire Green Hot Start II polymerase (ThermoFisher). The forward primer was phosphorylated at the 5′-end and its sequence was 5′-CAGGGCTCTCTAGATTGACT<u>TATGTATCCGCTCATGAGACAATAA</u>-3′, where underlined bases correspond to the annealing segment (and therefore to the internal primer for the subsequent partial re-synthesis) and the rest constitutes the final overhang. The reverse primers were functionalized at the 5′-end with three biotin or three digoxigenin molecules, to have asymmetric constructs. Products were cleaned using QIAquick PCR Purification Kit (Qiagen) and set to λ exonuclease digestion for 2 h at 37 °C, using 2 units of enzyme per μg of DNA. A heat treatment at 80 °C was then applied during 1 min to inactivate the exonuclease. The product was purified using 30 kDa Amicon centrifugal filters (Merck Millipore) and checked using agarose gel electrophoresis. If the digestion was successful, a linear PCR was performed on the ssDNA using 6 units Deep Vent (exo-), a polymerase that lacks 3′ → 5′ proofreading exonuclease activity, from New England Labs per μg of ssDNA. We used an initial denaturation step at 95 °C for 5 min, followed by 5 cycles of annealing (53 °C for 20 s), and extension (74 °C for 2.5 min),

and a final extension step at 74 °C for 5 min. The five cycles ensure that strand synthesis is complete and prevent mechanical failure of the tethers. The product was purified and concentrated to ~500 nM using a 50 kDa Amicon filter.

**Optical tweezers assay.** Carboxyl polystyrene beads (CP-20–10, diameter 2.1 μm, Spherotech) were covalently coated with sheep anti-digoxigenin antibody (Roche) via carbodiimide reaction (PolyLink Protein coupling kit, Polysciences, Inc.). Approximately 50 ng of the generated construct were incubated with 2 μL beads in 10 μL HMK buffer (50 mM HEPES, pH 7.5, 5 mM MgCl₂, 100 mM KCl) for 15 min in a rotary mixer at 4 °C and redituted in 350 μL HMK buffer. With our coupling strategy, ~50% of the constructs will be asymmetrically functionalized with digoxigenin and biotin in each side. To create the second connection, we employed Neutravidin-coated polystyrene beads (NVP-20–5, diameter 2.1 μm, Spherotech). Once trapped, beads were brought in close proximity to allow binding and tether formation was identified by an increase in force when the beads were moved apart. To mitigate photobleaching and tether damage, we added an oxygen scavenging system (3 units/mL pyranose oxidase, 90 units/mL catalase, and 50 mM glucose, all purchased from Sigma-Aldrich).

**Force spectroscopy data analysis.** Data were recorded at 500 Hz using a custom-built dual trap optical tweezers for the tether resistance assays and a C-Trap (Lumicks) for the dual monitoring experiments. Data were analyzed using custom scripts in Python. Optical traps were calibrated using the power spectrum of the Brownian motion of the trapped beads[65], obtaining average trap stiffness values of 0.39 ± 0.04 pN/nm. Force-extension curves were fitted to two worm-like chain models in series, using the approximation of an extensible polymer reported by Petrosyan[66] for the DNA, and the Odijk inextensible approximation for the protein contribution[67]. The contour length was 906 or 3500 nm for the two different DNA handles used (1.3 and 5 kb, respectively) and 120 nm for the MBP and 105 nm for YPet. Persistence length of the protein was fixed to 0.75 nm, whereas the persistence length and stretch modulus of the DNA handles were fitted and yielded average values of 30 nm and 700 pN/nm, respectively. Small persistence lengths do not necessarily reflect partial synthesis, as they are known to be considerably lower in the presence of multivalent ions[68], as the Mg2+ used in our measuring buffer and other recent optical tweezer studies[21,69].

Tether resistance was tested by slowly ramping up the tension on the tether and recording the rupture force. It is well known that the measured rupture force increases for higher pulling rates[70]. The pulling speed here used was 100 nm/s, too slow to bias the rupture force in any substantial way. If the DNA overstretching regime was reached, the tether was relaxed back. The rupturing force includes traces of the first pulls that showed proper MBP unfolding only (Supplementary Fig. 3a). For the lifetime experiments, the force was increased gradually to around 30 pN. The time between reaching this force and the rupture of the tether was recorded as the lifetime (Supplementary Fig. 3b). In Fig. 3d, the boxplots indicate the following: the median is displayed as a horizontal line within the box and the mean as a white square. Whiskers indicate the lowest datum still within 1.5 interquartile range (IQR) of the lower quartile and the highest datum still within 1.5 IQR of the upper quartile.

**Fluorescence imaging analysis.** For dual monitoring experiments, an excitation laser beam (with wavelengths of either 532 nm for YPet or 638 nm for Atto647N-trigger factor) was scanned along the beads and tether at a line rate of 12 Hz. The

excitation laser output power was 1.3 mW for YPet experiments and 0.3 mW for trigger factor binding experiments. Force spectroscopy and confocal microscopy data were synchronized based on the movement of the beads. The edge of the moving bead was tracked using a Gaussian fit and overlaid on top of the actual movement set in the mirror by minimizing the difference between the signals (Supplementary Fig. 9). This same movement was used to trace a region of the scanning between the beads including the protein (Supplementary Fig. 5b, red lines). Signal was calculated by adding the intensity of all pixels in that region and subtracting the background, calculated similarly by summing the intensity in a region of the same size outside of the beads.

We tested the emission lifetime of Atto647N under our experimental conditions using a labeled DNA construct (Supplementary Fig. 8a). Photobleaching was not observed in any of the confocal scanning experiments, which terminated upon tether rupture ($t_r = 660 \pm 150$ s, $N = 6$; Supplementary Fig. 8b). This timescale, which sets a lower limit for the dye lifetime, is much longer than the tens of seconds observed for trigger factor binding.

## Data availability
The data that support the findings of this study are available from the corresponding author upon reasonable request

## Code availability
Data were analyzed using a custom Python package that is available from the corresponding author upon reasonable request.

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

## Acknowledgements

We thank Günter Kramer for the labeled trigger factor samples. Work in the group of S.J. T. is supported by the Netherlands Organization for Scientific Research (NWO).

## Author contributions

D.P.M., M.J.A. and S.J.T. conceived the research. M.J.A., E.J.K. and D.P.M. developed and optimized the DNA–protein coupling assays. M.J.A., E.J.K., and V.S. purified the proteins. M.J.A. and E.J.K. performed the mechanical stability characterization. M.J.A. performed the simultaneous sensing–imaging experiments and the data analysis. M.J.A., D.P.M. and S.J.T. wrote the manuscript with input from all authors.

## Competing interests

The authors declare no competing interests.
