## [Peer Review File · Communications Chemistry]

Reviewers' comments:

Reviewer #1 (Remarks to the Author):

In this work, the authors developed a protein-DNA coupling strategy to covalently link a single protein domain to two long dsDNA handles at its two termini. Such construct can be mechanically manipulated using optical tweezers for investigations of the force-dependent stability of single protein domains. By integrating with fluorescence imaging, force-dependent interactions between the mechanical manipulated protein domain and other factors in solution can be studied. Overall, the strategy developed in this work is very useful for mechanical manipulation of single protein domains using optical tweezers. The manuscript is well written and easy to understand. The experimental data are convincing. I recommend the publication of this work in Communications Chemistry after a minor revision.

Page 2, line 27: The authors mentioned DNA overstretching transition but did not include a reference. The original discovery of this transition by Smith et al (Science, 271, 795–799) and Cluzel et al. (Science, 271, 792–794) should be referred.

Reviewer #2 (Remarks to the Author):

In this work, Avellaneda MJ et al. present a method to attach covalently long double-stranded DNA handles to proteins using Cys-Maleimide and YbbR-CoA chemical coupling. The method allows single protein manipulation and visualization with force spectroscopy methods like optical tweezers combined with confocal microscopy. This method may be of relative interest for the single-molecule manipulation community, specifically for studies of the mechanical folding/unfolding of proteins. However, the method seems not to provide any significant advance over other methods developed already for protein-DNA coupling (and properly referred by the authors) and probably, it will not influence thinking in the field. The authors did not develop new chemistry or alternative chemical modifications that may help experimentalist in the field to attach DNA to proteins. The method requires the inclusion of yppR tags or Cys at specific positions of the protein as reported already by other methods. Also, relatively high concentrations of highly purified proteins are required. The only relevant advance presented is the generation of ssDNA overhands through exonuclease degradation and subsequent PCR.

Overall, the work is well written and presented and the results are convincing.

Minor comments:

- Why a linear PCR reaction is required after exo degradation step? Why a PCR specifically? Why a single primer-extension DNA synthesis reaction would not be enough?
- Numbers 1, 2, 3, 4 in Figure 2a are hardly visible.

Typos:

- page 2, last paragraph of introduction, second line: sensing and imaging.. (two dots)
- page 6, line 3: several 100 nm.
- page 8, last paragraph: set to set to

Reviewer #3 (Remarks to the Author):

Avellaneda et al. present in this manuscript a new protocol to prepare DNA-handles for optical tweezers and covalently link their protein-of-interest to these handles. They present a protocol and

a characterization and two very brief examples how they applied their protocol.

Although it is always good to see new protocols arising, I am confused with this one. The authors have motivated their protocol mainly due to the effect that current protocols which generate the overhang by an abasic site cannot be covalently ligated easily. This is a very good point and some experiments suffer from this, many not. The presented approach is a special case, which could be also suitable for a more specialized journal. The chemistry in this manuscript is of limited novelty.

I have a few remarks which I would like to be addressed before I feel that this manuscript can be recommended for publication:

- Introduction: Page 1, last paragraph: The authors imply here that most studies were performed on DNA mechanics, but neglect the entire literature generated by the Rief lab or Cecconi lab, or Bustamante/Marqusee labs (protein folding) or protein degradation studies using ClpXP (Bustamante or Lang) I feel that this should not be ignored, but valued here that their protocol is an extension but not a complete new opener for a new field.
- Figure 3a: What are the two small arches on the DNA handle symbolize? covalent ligation?
- Figure 3c: what are the yellow dots?
- How do the authors ensure that Deep Vent products are completely double-stranded and not only partially synthesized? This is very important for handle mechanics which should be identical from molecule to molecule. Can you quantify this by mechanical or gel analysis? The authors see a persistence length of 30 nm. Why not 50 nm? Does this indicate partial synthesis of DNA handles?
- The authors imply in their introduction that no other methods for generating such handles exist. However, I am convinced that other methods exist, based e.g. on nicking enzymes generating DNA overhangs w/o basic sites which are perfectly ligatable. One example is given in Mukhortava et al NAR 2019. There it was used for DNA hairpin assembly, yet, it can readily be applied for protein-DNA coupling. I suggest the authors to perform a literature research regarding the nicking enzyme approach and include it in their introduction or discussion.
- I value the variety of examples presented as a proof of concept. Yet, I think the authors should discuss the results in more details. How does their Ypet data relate to Rief's GFP unfolding data? I am fully aware that this manuscript illustrates a new method for DNA tether formation for optical tweezers, yet, at the current state this manuscript appears rather thin. It lacks a rigorous comparison with existing protocols and the sample data is described very superficial. Can the authors showcase an example where experiments they perform were impossible before? It seems GFP unfolding or TF binding to unfolded proteins were previously possible.

Minor remarks:

- page 8, Methods: Please explain what SFP is and does? This is key for this protocol.
- page 10, the given laser power was at the objective, back focal plane? On which instrument was this measured? 1.3mW on a single fluorophore leads typically to instantaneous bleaching. Single-Molecule fluorescence experiments using confocal setups typically use 100-150 μ W back-focal plane powers. Please specify.

Replies to reviewers' comments

Reviewer #1 (Remarks to the Author):

In this work, the authors developed a protein-DNA coupling strategy to covalently link a single protein domain to two long dsDNA handles at its two termini. Such construct can be mechanically manipulated using optical tweezers for investigations of the force-dependent stability of single protein domains. By integrating with fluorescence imaging, force-dependent interactions between the mechanical manipulated protein domain and other factors in solution can be studied. Overall, the strategy developed in this work is very useful for mechanical manipulation of single protein domains using optical tweezers. The manuscript is well written and easy to understand. The experimental data are convincing. I recommend the publication of this work in Communications Chemistry after a minor revision.

We thank the referee for the kind words and the positive endorsement on our manuscript.

Page 2, line 27: The authors mentioned DNA overstretching transition but did not include a reference. The original discovery of this transition by Smith et al (Science, 271, 795–799) and Cluzel et al. (Science, 271, 792–794) should be referred.

Following this recommendation, we have added the references to the manuscript.

Reviewer #2 (Remarks to the Author):

In this work, Avellaneda MJ et al. present a method to attach covalently long double-stranded DNA handles to proteins using Cys-Maleimide and YbbR-CoA chemical coupling. The method allows single protein manipulation and visualization with force spectroscopy methods like optical tweezers combined with confocal microscopy. This method may be of relative interest for the single-molecule manipulation community, specifically for studies of the mechanical folding/unfolding of proteins.

We appreciate this acknowledgment.

However, the method seems not to provide any significant advance over other methods developed already for protein-DNA coupling (and properly referred by the authors) and probably, it will not influence thinking in the field.

We thank the referee for this comment. It also made us realize that we had not properly described the differences with existing methods. We have rewritten the introduction for this purpose.

We do agree that, in principle, it is not impossible to use existing methods for the new type of tweezers/fluorescence experiments we outlined on stable multi-protein complexes. In practice however, new methods are needed. In these types of experiments, coupling yield, handle length and handle strength are highly critical parameters. Current methods exhibit a limiting trade-off: it has not been possible to attach long and strong DNA handles with high efficiency. Direct covalent attachment of DNA tethers to proteins (Cecconi et al, 2004) does provide the required strength, but the coupling yield goes down dramatically with increasing DNA tether length, in particular for such dual-handle constructs. Hence, achieving the >3 kbp handle lengths that are required for fluorescence imaging becomes an impractical barrier. Secondly, proteins within complexes typically require forces >60 pN, as we showed previously for proteins interacting with the chaperone DnaK/Hsp70 or within a mini-aggregate (Mashaghi et al, Nature 2016, Ungelenk et al, Nat. Comm. 2016). Non-covalent two-step coupling methods based on DNA hybridization cannot sustain high forces for longer than a few milliseconds, and are therefore not suitable for these applications, despite their ability to generate long imaging-compatible handles with reasonable efficiency. Our method provides long and strong tethers with high coupling yields, which required a new combination of biochemical steps (see next point).

We stress that we consider here a new generation of tweezers/fluorescence experiments on tethered proteins, and the coming years will judge how important they will be. However, we believe that arguments for it are strong. There is a wealth of biology associated with conformational change in protein-protein complexes that has remained largely obscured. More generally, our coupling yield and strength improvements will be beneficial to a wider range of experiments, such as those on complex-to-purify proteins, which is relevant for human and disease-relevant questions.

The authors did not develop new chemistry or alternative chemical modifications that may help experimentalist in the field to attach DNA to proteins. The method requires the inclusion of yppR tags or Cys at specific positions of the protein as reported already by other methods.

We fully agree: the novelty of our approach does not concern new chemical modifications on the protein. The novelty rather lies in the way we produce the variable-size overhangs, which has not been used in single-molecule methods before, with a new combination of known enzymatic treatments. Indeed, a wide and versatile range of chemical modifications are

already available, including cysteine chemistry (Cecconi et al., European biophysics journal 2008), ybbR tags (Yin et al., PNAS 2005), sortase-mediated reactions (Koussa et al., Methods 2014) or click chemistry (Humenik et al., ChemBioChem 2007), among others. Most of these techniques are well-established, accessible and compatible with the protocol presented here (as we show with two examples). We believe this to be a strength of our approach: it is built upon and compatible with proven methods.

Also, relatively high concentrations of highly purified proteins are required.

Indeed, all existing protein-DNA coupling methods require initial relatively high concentrations of purified proteins, and ours is no exception. At the same time, we found that our high efficiency reduces the necessary material per experiment considerably. More specifically, we may estimate that our protocol requires approximately 40x less material than existing methods, based on the following. We obtain a 4x increase in the protein-anchor coupling yield compared to previous protocols using similar reactant concentrations and volumes (Mukhortava et al. Bioconjugate Chem. 2016), and a 10x increase for the attachment of DNA handles compared to the only study that reports this value to our knowledge, even when using a 3 times longer DNA handle (Fig. 2, Hao et al., Scientific Reports 2017).

The only relevant advance presented is the generation of ssDNA overhangs through exonuclease degradation and subsequent PCR.

We were not 100% sure what the reviewer meant here with 'the only relevant advance'. On the one hand we agree, as also discussed above. Indeed, the overhang generation (and subsequent ligation) are the novel biochemical steps in our protocol. But advances can be achieved in many ways. For instance, while comparisons are always difficult, the first seminal protein-DNA coupling method by Cecconi et al. also only used existing biochemical steps but did achieve a very important advance, we would argue. Here, we have identified the key current limitation, which is essentially the 3-way trade-off between efficiency, length, and strength. By enabling ligation, we could reduce the length of the anchors, increase coupling efficiency, and improve strength. We showed that varying the overhang length allows one to optimize this coupling efficiency. As mentioned earlier, we do believe that understanding and overcoming these technical challenges will be key in accessing an important new type of experiments.

Overall, the work is well written and presented and the results are convincing.

We thank the referee for the kind words.

Minor comments:

-Why a linear PCR reaction is required after exo degradation step? Why a PCR specifically? Why a single primer-extension DNA synthesis reaction would not be enough?

This is a good point, which has also helped us to spot a typo in the DNA overhang generation protocol. We perform 10 cycles of annealing and extension, but not DNA denaturation (see revised Methods). Single primer-extension poses the possible risk that the desired duplex DNA segment is re-synthesised only partially, which would affect the mechanical stability of the tethers (see reply to comment 6 of Referee 3). Therefore, we decided to perform several extension cycles to ensure complete strand filling, as shown in Fig. 2b. Motivated by this comment, we have revisited this step of the protocol, and we could not detect a substantial

difference between 1, 5 and 10 cycles (see Figure below). Given that there is no significant practical downside to using 5 cycles rather than 1 cycle, and for the reasons indicated above, we have decided to change the protocol description to 5 cycles, which reduces the duration to half.

-Numbers 1, 2, 3, 4 in Figure 2a are hardly visible.

This has been amended in the revised version of the manuscript.

Typos:

-page 2, last paragraph of introduction, second line: sensing and imaging.. (two dots)

-page 6, line 3: several 100 nm.

-page 8, last paragraph: set to set to

We thank the referee for their thorough and careful reading of the manuscript. The typos have been corrected in the revised version.

Reviewer #3 (Remarks to the Author):

Avellaneda et al. present in this manuscript a new protocol to prepare DNA-handles for optical tweezers and covalently link their protein-of-interest to these handles. They present a protocol and a characterization and two very brief examples how they applied their protocol.

Although it is always good to see new protocols arising, I am confused with this one. The authors have motivated their protocol mainly due to the effect that current protocols which generate the overhang by an abasic site cannot be covalently ligated easily. This is a very good point and some experiments suffer from this, many not. The presented approach is a special case, which could be also suitable for a more specialized journal.

We thank the referee for acknowledging the advantage of ligation-compatible handles. Reading back our text, we also realised that we should better describe our improvements. There are essentially two points:

1) The specific case we pursued here is the combination of tweezers with fluorescence, and ability to apply high forces. This may currently be considered as a special case. However, we argue that there is a strong rationale for it being a large application in the near future. There is a wealth of biology associated with conformational changes within protein-protein complexes that has remained largely obscured, and the tweezers-fluorescence combination will be more common. Protein-protein complexes require handle strength, as they typically have very stable folds, and strongly benefit from fluorescence, as protein binding-unbinding events need to be identified. We and others have thus far focussed on client-chaperone complexes, which is in itself a substantial field, but there are many other protein-protein interactions that are associated with important folding transitions and conformational changes. For instance, we have recently achieved the first measurements on disaggregase translocation with the coupling and tweezers-fluorescence strategy we present here, which would not have been feasible with the current methods.

2) Our method provides an advantage also for cases where fluorescence and/or high forces are not useful (e.g. for isolated protein (un)folding studies), namely its high coupling efficiency. This is the case with respect to direct handle-protein-handle coupling methods, and current two-step methods (first coupling of a small anchor, then DNA handle hybridization without ligation). The direct method is practically limited to very short handles of 500 bp. Even then the coupling efficiency is not high, and goes down dramatically with increasing handle length. For this reason, the two-step method was previously developed. Here, we show that decreasing the anchor length improves anchor-coupling efficiency, from 18% in previous studies (36% in our study) for the often used 34 nt to an efficiency of 85% for 20 nt (Fig. 1d). However, such 20 nt overhangs without ligation provides insufficient strength for most applications. This is also seen in figure 3, which shows tether rupture already at 25 pN, and lifetimes below 1 s at 30 pN. This is a reason many studies employ 34 nt anchors, as this length is required to provide sufficient strength – but thus at the expense of low coupling efficiency. For proteins that require high unfolding forces (close to 65 pN, like GFP), then even 34 nt anchors are not ideal, as the lifetime of the tethers is very short at such tensions.

Our method overcomes this tradeoff. Through ligation, it provides sufficient strength while still enabling high coupling efficiency. One may note that an efficient and strong coupling method is even more relevant for proteins that are complex to purify, as then there are other additional limiting factors that come into play.

The chemistry in this manuscript is of limited novelty.

On the one hand we agree. The novelty of our approach does not concern new chemical modifications on the protein. Indeed, the overhang generation (which has not been used in single-molecule methods before), and subsequent ligation, are the novel biochemical steps in our protocol. But advances can be achieved in many ways. For instance, while comparisons are always difficult, the first seminal protein-DNA coupling method by Cecconi et al. also only used existing biochemical steps, but did achieve a crucial advance that has spawned a new field, we would argue. Here, we have identified the key current limitation, which is essentially a trade-off between efficiency, length, and strength, and a way to overcome it.

I have a few remarks which I would like to be addressed before I feel that this manuscript can be recommended for publication:

- Introduction: Page 1, last paragraph: The authors imply here that most studies were performed on DNA mechanics, but neglect the entire literature generated by the Rief lab or Cecconi lab, or Bustamante/Marqusee labs (protein folding) or protein degradation studies using ClpXP (Bustamante or Lang) I feel that this should not be ignored, but valued here that their protocol is an extension but not a complete new opener for a new field.

The referee is right: many studies have shown the interaction between proteins using single-molecule force spectroscopy alone. Yet, to our knowledge this is the first time that such interaction has been studied using optical tweezers and fluorescence microscopy simultaneously. We did not mean to neglect the extensive literature on protein interaction studies, and following the recommendation of the referee we have added extensive background information to the introduction (page 2, second paragraph).

- Figure 3a: What are the two small arches on the DNA handle symbolize? covalent ligation?

We apologize for the lack of clarity. The two small arches on the DNA indeed represent T4 ligation. We have now added a description in the caption.

-Figure 3c: what are the yellow dots?

The yellow dots correspond to the non-ligated construct, but we agree that the different colours are redundant and we apologize for the confusion. We have changed the colour scheme of Figure 3 to improve clarity.

- How do the authors ensure that Deep Vent products are completely double-stranded and not only partially synthesized? This is very important for handle mechanics which should be identical from molecule to molecule. Can you quantify this by mechanical or gel analysis? The authors see a persistence length of 30 nm. Why not 50 nm? Does this indicate partial synthesis of DNA handles?

This is a very important point. Gel analysis of the product from the Deep Vent reaction revealed a complete re-synthesis of the complementary strand (Figure 2b). This was the case even with fewer extension cycles during the Deep Vent reaction, as evidenced by new experiments that we performed (see reply to referee 2, minor point 1). Small persistence lengths do not necessarily reflect partial synthesis, as they are known to be considerably lower in the presence of multivalent ions like Mg^{2+} (Baumann et al., PNAS 1997), as used in our measuring buffer and also reported in other recent optical tweezers studies (see Methods, Maillard et al.,

Cell 2011, Mukhortava et al., Bioconjugate Chem. 2016, or Bauer et al., PNAS 2018). This information has been added to the revised version of the manuscript (page 10, Methods, Force spectroscopy data analysis section).

- The authors imply in their introduction that no other methods for generating such handles exist. However, I am convinced that other methods exist, based e.g. on nicking enzymes generating DNA overhangs w/o basic sites which are perfectly ligatable. One example is given in Mukhortava et al NAR 2019. There it was used for DNA hairpin assembly, yet, it can readily be applied for protein-DNA coupling. I suggest the authors to perform a literature research regarding the nicking enzyme approach and include it in their introduction or discussion.

We thank the referee for pointing to alternative and interesting methods that can potentially be used to covalently couple DNA handles to proteins. We did not intend to imply that the method presented was the only route to ligation-compatible overhangs, and we have followed the advice to include nicking enzymes as a potential strategy in the discussion of the manuscript (page 8).

- I value the variety of examples presented as a proof of concept.

We thank the referee for the positive endorsement.

Yet, I think the authors should discuss the results in more details. How does their Ypet data relate to Rief's GFP unfolding data?

Following the recommendation, we have extended the discussion of the two proof-of-principle examples. The study mentioned by the referee (Ganim et al., PNAS 2017) is indeed a very relevant study. Our purpose was not to scientifically study the unfolding of a fluorescent protein, which would require a separate, more in-depth effort. Our purpose was rather a technical one: to highlight the importance of long, strong handles to enable simultaneous sensing and imaging of proteins. At this technical level, it may be of interest to discuss the following: in the Ganim study, the high unfolding forces of this GFP protein (47 pN) were applied only for a few tens of milliseconds to prevent handle rupture, as acknowledged in the Methods section of this manuscript. Our constructs can be held at those high forces for times at least 3 orders of magnitude longer (>20 s, Supplementary Fig. 4), thus making unfolding/refolding experiments more efficient.

Moreover, we note that experiments on tethered fluorescent proteins such as GFP are less sensitive to the handle length than with bound fluorescent proteins. In the former case, there are no labelled proteins in solution that represent a perturbing background signal and can bind to the beads and worsen their parasitic signal. The latter is illustrated by the trigger factor data.

I am fully aware that this manuscript illustrates a new method for DNA tether formation for optical tweezers, yet, at the current state this manuscript appears rather thin. It lacks a rigorous comparison with existing protocols and the sample data is described very superficial.

We understand this comment, as we had previously chosen to compare our method mainly to the two-step hybridization approach (Jahn et al., 2016), rather than to other previous protocols, as the latter were already treated more extensively in the literature. One difficulty also is that previous papers often do not report values for the obtained efficiencies (see below). Nevertheless, we have made an important effort to better highlight the advantages of the approach with rigorous experiments and solid data, as acknowledged by the other referees.

In general, our results demonstrate that the method resolves the existing three-way trade-off between coupling yield, handle length and strength. In particular:

- We more clearly outline that we have overcome efficiency limitations of previous DNA-protein coupling methods. Many two-step protocols employ 34 nucleotide-long DNA oligos for improved mechanical stability, and we could find only a few studies that report coupling yields. These are approximately 10-20% for two-handle attachments (Mukhortava et al., *Bioconjugated Chem.* 2017). Here, we improve the coupling yield to 85% by reducing the size of the DNA oligo (Fig. 1) without compromising mechanical stability.
- Similar technical coupling-methods papers do not report the efficiency of attaching the longer handles, with one exception to our knowledge (4% for 400 bp, Hao et al., *Scientific Reports* 2017). Here we achieved a 45% yield for the attachment of two 1300 basepair handles and 35% for 5000 basepair handles (Fig. 2).
- Additionally, here we have performed new experiments to show the yield of the covalent ligation. The principle of this test is that ligated constructs should remain intact by a heat treatment, while hybridized-yet-non-ligated constructs would not. Hence, we took a handle-protein-handle sample that was ligated, and a handle-protein-handle sample that was hybridized only. The samples were independently mixed with excess anchor and heated to 70 °C for 5 minutes and slowly cooled down to 16 °C (Supplementary Fig. 2a, see also below). While the majority of the ligated construct resisted the treatment, the non-ligated construct fully disassembled into single handles (Supplementary Fig. 2b, see also below).

- We consistently show the increased mechanical strength provided by covalent ligation with respect to non-covalent handles used in existing protocols (Fig. 3). We report not only higher sustained forces, but measurement durations several orders of magnitude longer, owing to the ligated handles.
- Finally, we have also compared the suitability for fluorescence of short and long handles. We show that the parasitic signal from the beads, more pronounced when labelled partners are added (trigger factor), can be eliminated by increasing the length of the DNA tethers to 5 kbp, almost twice the length previously used (Ganim et al., *PNAS* 2017). We find it remarkable that the attachment efficiency of such long handles remains high at 35% using our protocol.

Can the authors showcase an example where experiments they perform were impossible before? It seems GFP unfolding or TF binding to unfolded proteins were previously possible.

The referee poses a fair question, which has different aspects. First, we note that showing binding of a chaperones to a client at the single molecule level is technically challenging, and indeed has never been shown before. It is more challenging than the GFP (of which there is one study to date) or YPet experiments, because of the background signal of freely diffusing and labelled proteins. This challenge is related to the handle attachment that we developed here, as longer tethers are then needed. Hence, it has allowed this assay, which is the first proof-of-principle experiment of this kind.

Second, and in all fairness, we find it difficult to answer whether such TF binding experiments would have been possible with previous methods. A key point is that low efficiency can make the success rate of getting good data impractically low. While direct comparisons are difficult also because many yields are not reported, the available data suggests that direct handle coupling methods would have been impossible here, as it only works for very short tethers. Previously used hybridization-only methods with 34 nt anchors suggest a yield of ~18% double-anchor-coupling efficiency (vs 85% here). Given the long handles needed here due to the background signal, which we stress were previously not as crucial, it is unclear what the full handle-protein-handle yield would have been, and if it would have been sufficient. But clearly the efficiency of obtaining a proper tether would have been substantially lower. Note that one must also take the limited TF-labelling efficiency and required low TF concentration into account, which further reduces the chance of seeing binding. With these additional uncertainties, it is key to have tethers that are long yet are established efficiently. With proteins that are more complex to purify, one encounters further reductions in the success rate of experiments. Our general rule of thumb is that if one cannot obtain one good experiment per day, then the reproducibility is too low for a proper scientific study, though this may vary for different types of study.

Perhaps we can here again mention a recent example from our own lab, where we measured the translocation activity of a disaggregase for the first time. Here long handles were essential, because fluorescence showed where the disaggregase was positioned, and high stability was essential, because the motor applied forces over 65 pN. This is one new example of the need for this coupling approach. In addition, we routinely find very high stabilities in client-chaperone complexes, which likely are present in other protein-protein complexes and large proteins as well.

Overall, we are convinced that our method will be of direct use to many of our colleagues. While for many applications current methods certainly are sufficient, as evidenced by the many beautiful papers that our field has produced, we do believe that our method will be very helpful in many existing and new applications, whether they involve fluorescence imaging or not.

Minor remarks:

- page 8, Methods: Please explain what SFP is and does? This is key for this protocol.

In addition to the description in the main text and the schemes of Supplementary Fig. 1, we have added more details to the Methods (page 9).

- page 10, the given laser power was at the objective, back focal plane? On which instrument was this measured? 1.3mW on a single fluorophore leads typically to instantaneous bleaching. Single-Molecule fluorescence experiments using confocal setups typically use 100-150 μ W back-focal plane powers. Please specify.

We use a commercial C-trap setup from Lumicks, as specified in the Methods. The power reported there is the set laser output power, which is attenuated through the optics of the instrument. Unfortunately, we thus cannot report this attenuation factor nor the laser power at the sample plane.

REVIEWERS' COMMENTS:

Reviewer #2 (Remarks to the Author):

Avellaneda et al. describe a new method to attach covalently long double-stranded DNA handles to proteins. The author's rebuttal letter and the modifications included in the new (revised) manuscript address adequately the issues raised before, and clarify the novelty of their method, which was not evident in the previous version. The new method provides significant advance over other methods previously described, as it provides higher yields and stabilities. The method seems to be particularly suitable to perform mechanical-fluorescence studies and may be of special relevance in this field. The work is well written and results are convincing.

Main points:

I would suggest the authors discuss briefly the main limitations of their method.

For example, working with DNA oligos bearing a single digoxigenin at the end will limit significantly the strength of the attachments. Interestingly, using a single digoxigenin at one end, the authors measured over-stretching transitions in 71% of the pulling events. This is a pretty high yield, considering the mechanical strength of individual dig-antidig connections. These data may indicate that they used high pulling rates. What is the pulling rate they used?

Low pulling rates may be required to identify unfolding intermediates in some proteins. In such cases, a 'stronger' or more mechanically stable chemistry than that of a single dig-antidig connection may be needed. The work will benefit from the inclusion of alternative proposals to overcome this issue. What other chemistry compatible with their method could be used to overcome this problem?

Also, the authors should indicate explicitly that their method is useful to detect protein unfolding/folding transitions below the overstretching force of DNA (<60 pN). DNA overstretching transition will complicate the identification of unfolding transitions occurring around/above 60-70 pN.

Reviewer #3 (Remarks to the Author):

Avellaneda present here a revised manuscript for their method of protein coupling to DNA for optical tweezers experiments. They have addressed most of my comments very well. Even though they present here a new collection of methods to achieve their goal of higher efficiency for protein-DNA conjugates, I am still skeptical about the overall big step forward. Let me mention two points here, which I find important to consider: (i) The authors argue the way of novelty with referring to Cecconi et al, however, Cecconi used and introduced this method within a bigger context of a biological study of protein folding with optical tweezers (Science 2005) and only wrote a small technical paper in European Biophysical Journal afterwards. Their method was 15 years ago indeed novel and allowed for the first time protein-DNA conjugates to be stretched by optical tweezers. This is not the case here. (ii) The authors argue that their conjugation work allows first time such trigger factor studies. Yet, there are various examples in literature studying trigger factor activity on protein folding using force spectroscopy, like Liu...Kaiser, Mol Cell 2019 and Haldar...Fernandez, Nat Comms 2018. I feel the authors line of arguments is not solid enough.

Despite these points, I agree that it might be useful to publicise this new protocol of sample design.

Replies to reviewers' comments

Reviewer #2 (Remarks to the Author):

Avellaneda et al. describe a new method to attach covalently long double-stranded DNA handles to proteins. The author's rebuttal letter and the modifications included in the new (revised) manuscript address adequately the issues raised before, and clarify the novelty of their method, which was not evident in the previous version. The new method provides significant advance over other methods previously described, as it provides higher yields and stabilities. The method seems to be particularly suitable to perform mechanical-fluorescence studies and may be of special relevance in this field. The work is well written and results are convincing.

We thank the referee for the kind words and the positive endorsement of our manuscript.

Main points:

I would suggest the authors discuss briefly the main limitations of their method.

For example, working with DNA oligos bearing a single digoxigenin at the end will limit significantly the strength of the attachments. Interestingly, using a single digoxigenin at one end, the authors measured over-stretching transitions in 71% of the pulling events. This is a pretty high yield, considering the mechanical strength of individual dig-antidig connections.

We agree. For this reason, our oligos bear three digoxigenin molecules rather than one, and hence provide higher mechanical stability. Similarly, we use three biotins on the other end of the construct. We have edited the methods section to make this clearer.

These data may indicate that they used high pulling rates. What is the pulling rate they used? Low pulling rates may be required to identify unfolding intermediates in some proteins. In such cases, a 'stronger' or more mechanically stable chemistry than that of a single dig-antidig connection may be needed. The work will benefit from the inclusion of alternative proposals to overcome this issue. What other chemistry compatible with their method could be used to overcome this problem?

The pulling speed here used was 100 nm/s. This is a slow rate well suited to identify possible folding intermediate states that the reviewer refers to. As mentioned above, we do use constructs with more than one dig-antidig connection to provide such stronger attachments.

We also note that the referee might have missed this relevant description from the methods (page 11, second paragraph): "Tether resistance was tested by slowly ramping up the tension on the tether and recording the rupture force. It is well known that the measured rupture force increases for higher pulling rates. The pulling speed here used was 100 nm/s, too slow to bias the rupture force in any substantial way."

We have now also included additional discussion on alternative linkages that can be integrated in our modular approach and may become useful when even longer lifetimes at high forces are required (page 6).

Also, the authors should indicate explicitly that their method is useful to detect protein unfolding/folding transitions below the overstretching force of DNA (<60 pN). DNA overstretching transition will complicate the identification of unfolding transitions occurring around/above 60-70 pN.

We agree, and have made the suggested changes (page 6). Note that unfolding >60pN can be useful, even without observing the unfolding transitions, in order to obtain a stretched polypeptide as the starting state.

Reviewer #3 (Remarks to the Author):

Avellaneda present here a revised manuscript for their method of protein coupling to DNA for optical tweezers experiments. They have addressed most of my comments very well.

We thank the referee for the kind words.

Even though they present here a new collection of methods to achieve their goal of higher efficiency for protein-DNA conjugates, I am still skeptical about the overall big step forward. Let me mention two points here, which I find important to consider: (i) The authors argue the way of novelty with referring to Cecconi et al, however, Cecconi used and introduced this method within a bigger context of a biological study of protein folding with optical tweezers (Science 2005) and only wrote a small technical paper in European Biophysical Journal afterwards. Their method was 15 years ago indeed novel and allowed for the first time protein-DNA conjugates to be stretched by optical tweezers. This is not the case here. (ii) The authors argue that their conjugation work allows first time such trigger factor studies. Yet, there are various examples in literature studying trigger factor activity on protein folding using force spectroscopy, like Liu...Kaiser, Mol Cell 2019 and Halder...Fernandez, Nat Comms 2018. I feel the authors line of arguments is not solid enough.

It is not fully clear to us what the reviewer argues here.

(i) The mentioned high-impact Science article and the technical paper, which was smaller but also heavily cited, only convey how much that method was appreciated. To us this therefore strengthens the point we were making, as this success was despite the fact that their method combined existing biochemical steps.

(ii) regarding existing work: We clearly were not claiming that our paper is the first force spectroscopy work on trigger factor, as the reviewer seems to suggest, but rather to combine it with fluorescence detection of single trigger factor (un)binding events, which these papers do not consider. If the point is to cite these papers; that is a good idea because they do help sketch the wider scientific landscape. Hence, we have now cited them in the introduction of the revised manuscript.

Despite these points, I agree that it might be useful to publicise this new protocol of sample design.

We thank the referee for the positive endorsement of our manuscript.